# Diminished circadian and ultradian rhythms of human brain activity in pathological tissue in vivo

Christopher Thornton[1,4], Mariella Panagiotopoulou[1,4], Fahmida A. Chowdhury[2], Beate Diehl[2], John S. Duncan [2], Sarah J. Gascoigne [1], Guillermo Besne[1], Andrew W. McEvoy[2], Anna Miserocchi[2], Billy C. Smith[1], Jane de Tisi[2], Peter N. Taylor[1,2,3] & Yujiang Wang [1,2,3] ✉

Chronobiological rhythms, such as the circadian rhythm, have long been linked to neurological disorders, but it is currently unknown how pathological processes affect the expression of biological rhythms in the brain. Here, we use the unique opportunity of long-term, continuous intracranially recorded EEG from 38 patients (totalling 6338 hours) to delineate circadian (daily) and ultradian (minute to hourly) rhythms in different brain regions. We show that functional circadian and ultradian rhythms are diminished in pathological tissue, independent of regional variations. We further demonstrate that these diminished rhythms are persistent in time, regardless of load or occurrence of pathological events. These findings provide evidence that brain pathology is functionally associated with persistently diminished chronobiological rhythms in vivo in humans, independent of regional variations or pathological events. Future work interacting with, and restoring, these modulatory chronobiological rhythms may allow for novel therapies.

Physiological processes are often modulated and structured by chronobiological rhythms. The daily, or circadian rhythm is perhaps the most pertinent and well-studied rhythm with a central pacemaker in the suprachiasmatic nucleus. Such rhythms are also differentially expressed in a range of tissues and organs, entrained by the central pacemaker. For example, multiple organs have been shown to display their own circadian rhythm in terms of gene expression and translation even in the absence of the pacemaker[1]. Even within a single organ, such as the brain, circadian regulation of gene expression is tissue-specific[2,3]. This local, or tissue-level, autonomous rhythmicity allows tissue-specific adaptations in phase or magnitude[4], whilst being coordinated by a central pacemaker. Other chronobiological rhythms also exist on longer and shorter timescales, termed infradian and ultradian rhythms, respectively. However, these are far less well-studied, appear to differ in magnitude and period between individuals, organs and

tissues, and their biological mechanisms are elusive. So far, candidate drivers of some ultradian rhythms have been proposed, such as the pulsatile cortisol secretion in mammals every 1–3 hours[5,6], where various organs and tissues may react differently[7,8]. Taken together, there is evidence that, with or without a central pacemaker, circadian and ultradian rhythms are differentially expressed at the tissue level to temporally structure and modulate local physiological processes.

Disrupted chronobiological rhythms are often associated with dysfunction and disease. Associations between altered daily/circadian rhythms and neurodevelopmental disorders, mood disorders, epilepsy, Parkinson's and dementia have been reported[6,9]. Similarly, disrupted ultradian rhythms may also be associated with neurological disorders, although their underpinning mechanisms are only beginning to be explored in animal models[10,11]. Many of the reported associations are linked to behavioural disruptions on circadian and

[1]CNNP Lab, School of Computing, Newcastle University, Newcastle upon Tyne, UK. [2]UCL Queen Square Institute of Neurology, London, UK. [3]Faculty of Medical Sciences, Newcastle University, Newcastle upon Tyne, UK. [4]These authors contributed equally: Christopher Thornton, Mariella Panagiotopoulou. ✉e-mail: Yujiang.Wang@newcastle.ac.uk

ultradian timescales (e.g., sleep, levels of physical activity). However, tissue level regulation and expression of chronobiological rhythms in brain activity are less well studied. One ex-vivo study with 16 patients with drug-refractory temporal lobe epilepsy showed that the expression of the circadian clock gene *Bmal1* is reduced in surgically resected pathological tissue compared to healthy tissue[12]. Similarly, tissue-level disruption to clock gene transcription and translation are found in various animal models of epilepsy[13].

Thus, emerging evidence hints at tissue-level alteration of chronobiological rhythms in pathological tissue in brain disorders. However, to date, it is completely unknown if pathological tissue shows different expression of chronobiological rhythms in brain activity and electrophysiology in vivo in humans.

Here, we therefore investigate the rhythmicity of human brain activity in vivo. EEG is a commonly used method to measure brain activity patterns and multiple studies have shown that rhythms on circadian and multiple ultradian timescales can also be captured as the rhythmic modulation of particular signal properties (e.g. refs. [14–19]). This includes clear evidence of rhythms in the canonical EEG frequency bands following a circadian timescale[20]. In contrast to, e.g., gene expression data, EEG data has the clear advantage that it can be sampled continuously at a very high rate, allowing for the application of robust time series methods, such as Fourier analysis[21], wavelet analysis[22], and empirical mode decomposition[19]. In this study, we will use intracranially recorded continuous EEG (iEEG) over multiple days to measure circadian and ultradian rhythms in humans in vivo. Intracranial recordings have the advantage of a high signal-to-noise ratio, and provide excellent spatial resolution, as the electrodes directly sample from the target brain tissue without interference. This allows us to delineate functional chronobiological rhythms of brain activity in specific locations.

In the following sections we will examine (1) if circadian and ultradian rhythms are altered in pathological tissue, (2) if the alterations can be attributed to pathology alone and (3) if any alterations are dependent on seizure occurrence.

## Results
### Overview of data and analysis
We retrospectively analysed long-term iEEG recordings from 38 individuals with refractory focal epilepsy from the National Hospital for Neurology and Neurosurgery (Table 1). These recordings were performed as part of the pre-surgical evaluation in these patients, and this retrospective study does not explicitly account for environmental factors/drivers of ultradian and circadian rhythms. Therefore, we only use the terms ultradian and circadian here to refer to the timescale of the rhythms we observe in the iEEG data, and do not imply any environmental conditions with these terms.

From the iEEG recordings of each patient, we localised each recording contact to an anatomical region of interest (ROI) and calculated the relative band power in the delta range (1–4 Hz) and the other four frequency bands (theta : 4–8 Hz, alpha : 8–13 Hz, beta : 13–30 Hz and gamma : 30–47.5 Hz, 52.5–57.5 Hz, 62.5–77.5 Hz) using a non-overlapping 30 s window over the continuous recording in all patients (recording duration between 2-21 days). Circadian (a period of 19–31 hours) and ultradian (1–3 h, 3–6 h, 6–9 h, 9–12 h, and 12–19 h periods) rhythms were then isolated by applying band pass filters to the signal of relative band power over days in each ROI.

Clinically identified regions where seizures originate (seizure onset zone, or SOZ) are often used to approximate areas that are hypothesised to cause or support seizures as part of a network. These regions are often also associated with some histopathological finding when resected. We therefore propose to use SOZ regions as a proxy for pathological tissue. To ensure robustness of our results, we also used the alternative definition of the tissue that was subsequently surgically removed, as it was deemed central to generating epileptic seizures. We obtained similar results (Suppl. S5) with either definition, and will use the broad term of pathological tissue throughout the paper to refer to tissue that is most likely functionally (causing seizures) and/or structurally pathological.

### Circadian rhythms of brain activity are diminished in pathological tissue
Figure 1a shows the power of the delta EEG band over seven days for an example patient, filtered to isolate only the circadian rhythm (19h-31h). In this patient the regions with the weakest circadian rhythm are within the left inferior parietal lobe, visualised in Fig. 1b. These comprise two out of the three pathological regions. Figure 1c compares the power of the delta circadian rhythm in regions with pathology against the remaining regions. We quantify this difference using the non-parametric Area Under the Curve (AUC) of a Receiver Operating Characteristics analysis[23]. Here, AUC values >0.5 imply diminished power in pathological tissue and the example patient yielded an AUC = 0.92. Figure 1d shows the AUC values across all 38 patients - the distribution is substantially >0.5 (median AUC = 0.62, $p = 0.005$) implying a diminished circadian rhythm in delta power in the pathological tissue across a majority of patients (66% of subjects with AUC > 0.5). When we perform the same analysis for the other canonical EEG bands (Theta, Alpha, Beta, Gamma) we find similar results (supplementary table S2). We did not find noteworthy modulatory effects when investigating the role of age, sex, anti-seizure medication and epilepsy type. These results are shown in Supplementary Figs. S7.1 and S7.2. Applying a more narrow definition of circadian rhythms (20–26 h) did not substantially alter the distribution of AUCs (median AUC = 0.6) – shown in Supplementary Fig. S8.1. In the example subject, we see lower cycle amplitude at the start and end of the recording, a feature present in a minority of cases across the cohort. Patterns of cycle amplitude across the recording are explored in supplementary section S11.

### Multiple ultradian rhythms are diminished in pathological brain tissue
Figure 2a shows the delta band power, filtered to isolate the 3–6 h ultradian rhythm, across regions for the same example patient. As with the circadian rhythm, there was a diminished power in the three pathological regions, quantified with an AUC of 0.96 for this patient and visualised in Fig. 2b. Diminished ultradian rhythms are present across the cohort, with a median AUC of 0.69 for the 3–6 h rhythm, and median AUC values above 0.5 for all rhythm periods - shown in Fig. 2c, $p < 0.05$ for all. When we perform the same analysis for the other canonical EEG bands (Theta, Alpha, Beta and Gamma) we again find similar results (supplementary table S2).

In Suppl. S4, we further show that these effects in circadian and ultradian rhythms are not simply due to a diminished magnitude of the raw EEG signal in the pathological tissue.

### Pathology is independently associated with diminished chronobiological rhythms
To investigate if our observed effects are potentially explained by the spatial location, or the broad brain lobe (or subcortical region) of the

## Table 1 | Summary of patient data used in the analysis

| N | 38 |
|---|---|
| Age (mean, SD) | 32.1 (8.5) |
| Sex (M, F) | 17, 21 |
| Temporal, extratemporal | 17, 21 |
| Side (left, right) | 22, 16 |
| Num contacts (mean, sd) | 72.8 (25.2) |
| Recording duration in hours (mean, sd) | 111.5 (58.1) |

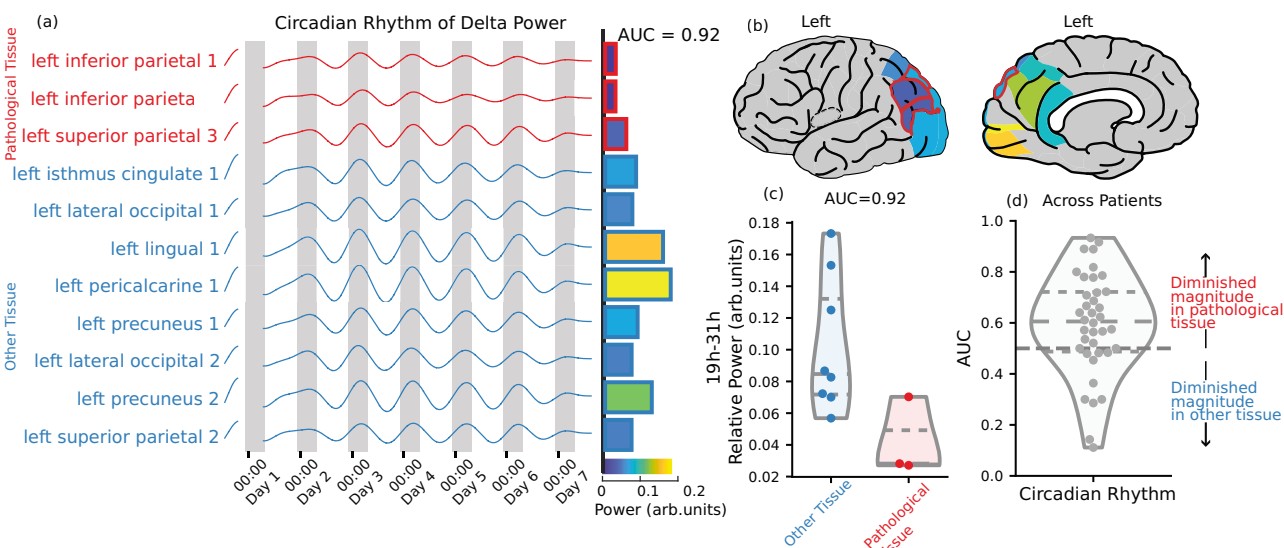

**Fig. 1 | Circadian rhythm of delta power is diminished in brain areas with pathology. a–c** From subject A: (**a**) The circadian rhythm of delta power, obtained by applying a band-pass filter to isolate the power of the signal with a period between 19 and 31 hours. Pathological tissue is shown in red. Bars indicate the relative power of the signal in each ROI. **b** The relative power in each brain area, pathological tissue is outlined in red. **c** The relative power of the circadian rhythm in pathological tissue compared to other tissue. An AUC of 0.92 is calculated when using this relative power to predict pathology. **d** The AUC of each subject for the same prediction. An AUC > 0.5 indicates that the rhythm is diminished in pathological tissue. Dashed lines indicate median and quartiles of the distribution. Brain plots from ref. 50. Source data are provided as a Source Data file.

observations, we plotted the power of an example ultradian rhythm in delta grouped by lobe across patients (Fig. 3a). While there is some variation of power of this ultradian rhythm across lobes, the power in the pathological tissue in each area appears consistently diminished.

To show that pathology is independently associated with diminished chronobiological rhythms, we tested the association in a mixed effects regression with the lobe (those shown in Fig. 3a) as a covariate. This model took the power of the rhythm in each ROI as the dependent variable, the lobe of the ROI and whether the ROI has pathology (is within the SOZ) as independent variables, and grouped by patient. Figure 3b shows the partially standardised fixed effect for pathology ($\beta_1$) for each ultradian/circadian rhythm in delta band power. All coefficients are below zero, indicating that ultradian/circadian rhythms are diminished in pathological tissue. Applying a likelihood ratio test to compare a model with pathology and brain area to a model with brain area only, we find that the effect of pathology explains the power of cycles significantly better for almost all EEG bands and ultradian/circadian rhythms (see Suppl. S2 for details).

Finally, we also checked if any particular type of pathology was associated with diminished rhythmicity, and found that most subjects showed this effect without evidence for the influence of the type of pathology (Suppl. S6).

**Diminished rhythms are persistent in time and independent of seizure occurrence**

To assess whether the rhythms of brain activity remained diminished in pathological tissue consistently over time and not only during the period around epileptic seizures (peri-ictal period), we calculated a rolling median of the AUC using a window proportional to the rhythm period multiplied by 1.5 (e.g., 36 h for the circadian rhythm). We illustrate the circadian rhythm of delta power in our sample patient in Fig. 4a, the rolling AUC is shown in Fig. 4b along with a histogram showing that the distributions of AUC values are similar during inter-ictal and peri-ictal periods. Figure 4c aggregates results across subjects and rhythms, showing no difference in the AUC distributions of inter-ictal and peri-ictal periods. The overall seizure load (average number of seizures per hour) was also not correlated with the AUC for the

circadian rhythm of delta ($r = -0.04$) and no strong correlation was found for any other rhythm ($r < 0.3$, see Suppl. S3). A similar analysis of only the post-ictal period found similar AUC distributions (supplementary section S10 and the phase preference of seizures across the cohort is characterised in supplementary section S9.

## Discussion

Analysing long-term iEEG recordings from 38 subjects with refractory focal epilepsy, we identified rhythms in the power of canonical EEG bands at circadian and ultradian timescales. We found that these rhythms were diminished in pathological tissue, and that this association remained when we controlled for brain area. We also found that these rhythms remain diminished persistently across time, without dependence on seizure occurrence. Our work provides initial evidence of an association between the strength of chronobiological rhythms and the healthy functioning of brain tissue in humans in vivo.

Circadian clock gene dysfunction has been seen in pathological tissue in multiple animal models[13] and human tissue[12], but a key advance in our study is to also demonstrate a functional abnormality in pathological tissue in terms of electrophysiological function in humans. From animal models we have learned that circadian gene oscillation patterns do not necessarily translate to similar changes in mRNA or protein level[13]. It was therefore also unclear if disruptions in these patterns translate to electrophysiological dysfunction, particularly in humans. Although numerous studies have reported circadian and ultradian rhythms in a range of signal properties in human EEG, these have been reported both in healthy participants[20,24,25], as well as in e.g. epilepsy patients[19,26]. It was therefore unknown if (i) there was a dysfunction in circadian and ultradian rhythms in electrophysiology associated with pathology in humans and (ii) if so, how the dysfunction would be expressed. Our work therefore serves as crucial evidence that (i) dysfunction is seen in pathology in circadian and ultradian rhythms in electrophysiological function, and (ii) dysfunction is expressed as a persistently diminished rhythm.

Electrophysiological function is, of course, not limited to the signal features of relative band power that we analysed. Indeed, there is evidence that other EEG signal features may display different

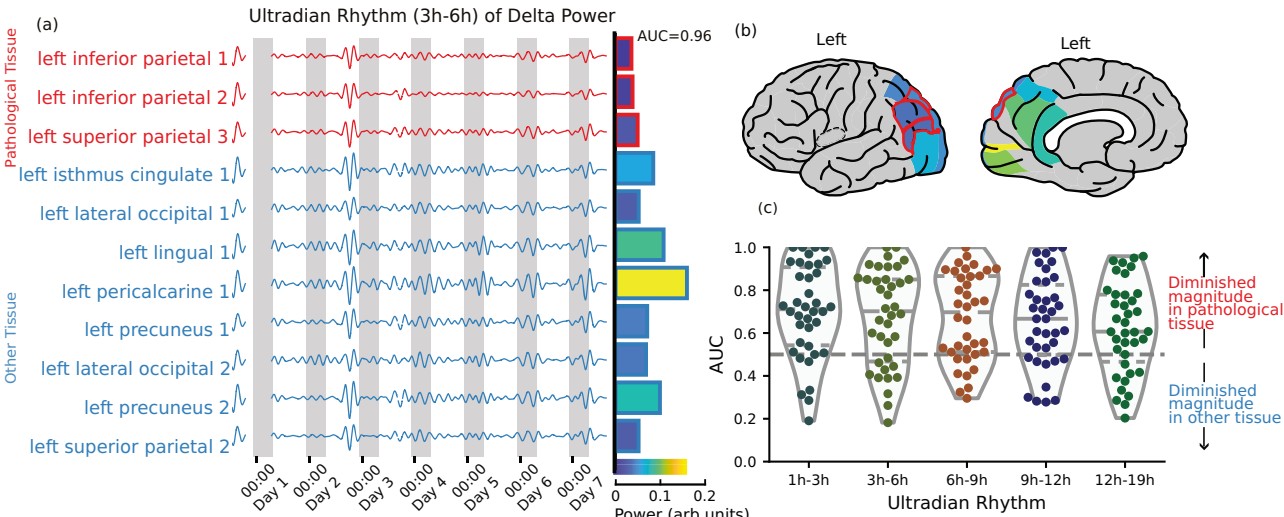

**Fig. 2 | Ultradian rhythm of delta power is diminished in brain areas with pathology. a–c** From subject A: (**a**) An ultradian rhythm of delta power, obtained by applying a band-pass filter to isolate the power of the signal with a period between 3 and 6 hours. Pathological tissue is shown in red. Bars indicate the relative power of the signal in each ROI. **b** The relative power of the ultradian rhythm (3–6 h) in each brain area, pathological tissue is outlined in red, no recording was performed in grey areas. **c** The AUC of each subject when predicting pathology from the power of each ultradian rhythm. An AUC > 0.5 indicates that the rhythm is diminished in pathological tissue. Dashed lines indicate median and quartiles of the distribution. Brain plots from ref. 50. Source data are provided as a Source Data file.

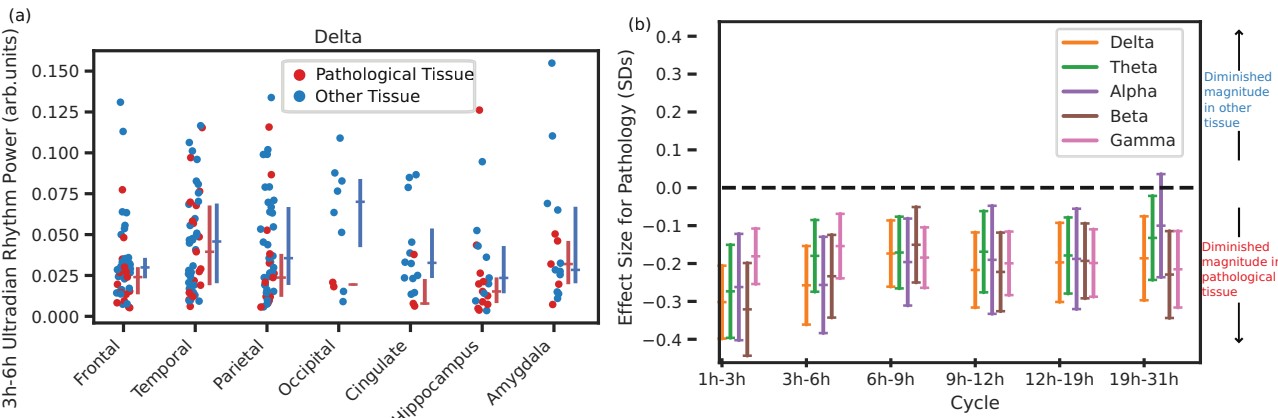

**Fig. 3 | Tissue pathology is associated with diminished rhythms while controlling for brain region. a** The power of the delta ultradian rhythm (3–6 h) in each lobe or region for all patients, with each dot showing the power of the rhythm recorded in that region – some patients have multiple electrodes per region. Red dots indicate electrodes recording from pathological tissue, blue from other tissue. The lines indicate the median and inter-quartile range. These are calculated by first calculating a median power for each individual in each region (if they have more than one electrode in that region) and then we calculate a population median and interquartile range across all patients who have electrodes within the region. **b** The effect size (partially standardised coefficient and 95% confidence interval) for tissue pathology in a mixed effects regression relating brain region and tissue pathology to rhythm power. Shows the expected difference in rhythm power (in standard deviations) for pathological tissue, while accounting for variability across regions and individuals. The centre line indicates the estimated effect size, the error bars indicate the 95% confidence interval. Source data are provided as a Source Data file.

rhythms[26]. Future work should investigate a fuller range of signal features and seek to answer questions such as whether different pathologies display different profiles of disruption in rhythms. Nevertheless, relative band power is a primary set of features that is both widely accepted and used in neuroscience, making it easier to compare and put our results into context. We are also encouraged that our main results replicate over all frequency bands. Finally, our results are in agreement with a wider literature: diminished rhythmicity appears to be consistently observed in multiple modalities and organs in ageing and pathology[27,28].

Another open question is if the electrophysiological circadian disruptions are a cause or consequence of the epileptogenic pathology. In our data, we observed no correlation between the resection of brain areas showing diminished rhythms and subsequent surgical outcome. This observation would suggest that sparing regions with diminished rhythms does not cause recurrent seizures after surgery per se. Similarly, we observed no temporal correlation between the magnitude of diminished rhythms and seizure occurrence. These observations should be validated in a larger cohort in future, ideally with more complete spatial sampling. We can nevertheless conclude that we see no evidence of disrupted rhythms causing seizures directly or acutely in our cohort. This however, does not rule out indirect effects of disrupted rhythms over longer timescale, where disrupted rhythms impact sleep or other restorative processes over days and weeks and as a consequence provoke or at least enable seizures.

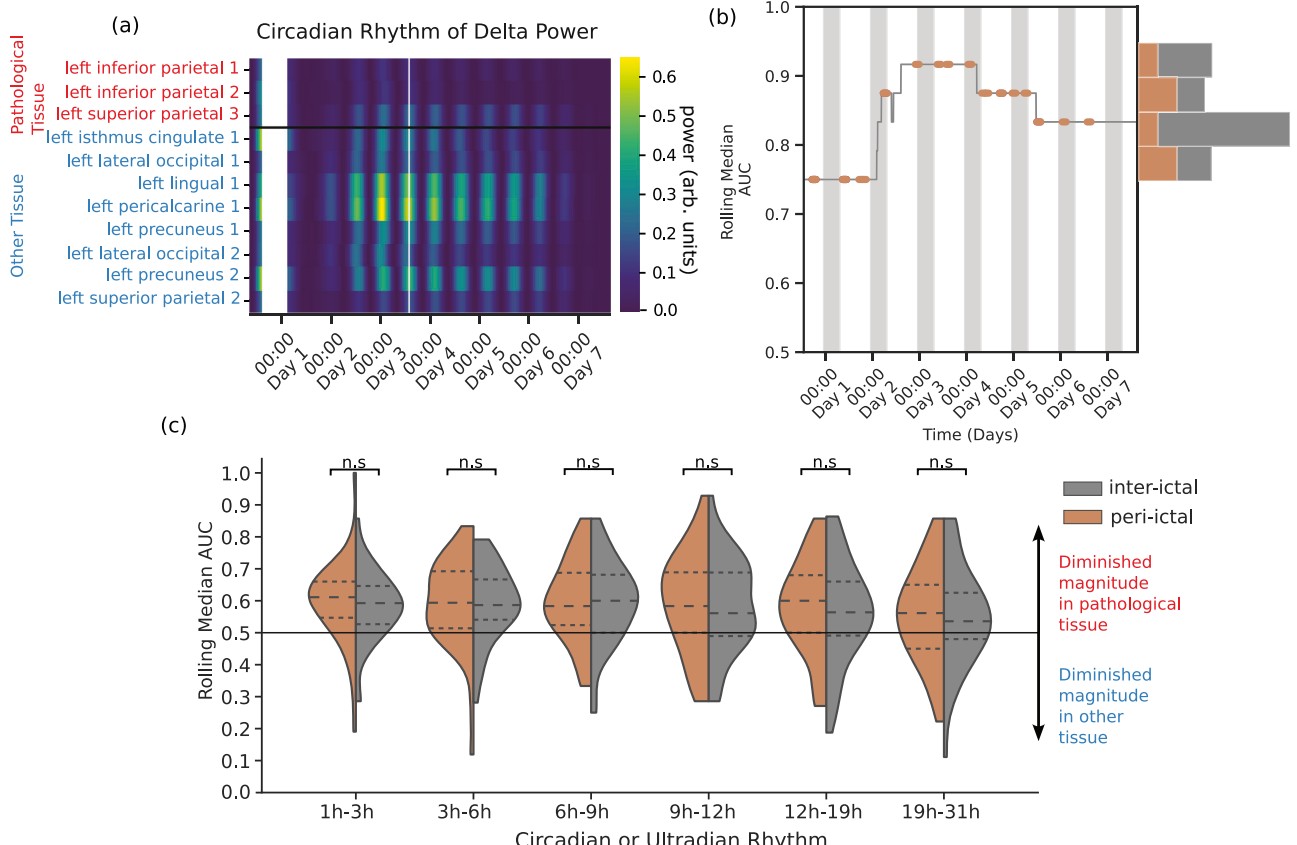

**Fig. 4 | Rhythms remain diminished in pathological tissue across time and independent of seizure occurrence. a–c** Example subject A: (**a**) The power of the circadian rhythm of delta across ROIs. Each cell represents the power of the circadian rhythm in the corresponding ROI and 30 s time segment. ROIs in red are deemed pathological. **b** The median AUC of the circadian rhythm captured using a rolling window of 36 hours. The peri-ictal period (1 hour before seizure start to 1 hour after its end) is shown in orange. The histogram shows the distribution of AUC values calculated during peri-ictal and inter-ictal periods. **c** Compares rolling AUC values calculated during inter-ictal and peri-ictal periods for each circadian and ultradian rhythm of delta power. A two-sided paired Student's $t$ test, with no correction for multiple comparisons indicates no difference between periods for each rhythm. Dashed lines indicate median and quartiles of the distribution. Source data are provided as a Source Data file.

Interestingly, circadian and multi-day rhythms have attracted great interest recently: Various EEG and physiological signals display circadian and multi-day rhythms, and seizures occurrence[29–31], as well as other seizure features[19,32,33] couple to the phase of these rhythms in most patients. However, the mechanism of this coupling remains elusive, and the key question is *how* the seizure-generating process was modulated. Our recordings were too short for most patients to investigate multi-day rhythms, but our work can contribute towards the circadian aspect of the question. Our data suggests the epileptogenic pathology actually experiences a persistently diminished circadian modulation, most likely caused by the pathology itself. Therefore, we suggest that the circadian modulation of seizure occurrence and other seizure features is most likely mediated as a network phenomenon with other non-pathological regions, rather than arising locally within the pathological tissue itself. This supposition is also supported by recent observations that interictal markers of epilepsy are also more persistent, and less rhythmic in pathological tissue, possibly independent of seizure occurrence[34,35]. Future work will investigate if direct evidence of this network modulation can be found.

In general, a bi-directional relationship is often highlighted between circadian rhythmicity and neurological conditions[9], particularly epilepsy[36]. A vicious circle is often described, where pathology or pathological events may erode or disrupt circadian rhythms, and disrupted rhythms may in turn exacerbate disease symptoms or even support disease progression. A key component in this vicious circle is sleep, which we have not investigated explicitly in our study. However, future work should consider if alterations in both ultradian and circadian rhythms may reflect a loss of healthy sleep physiology regionally and behaviourally. Much of this complex interplay between rhythms, sleep, and disease is still unknown; nevertheless, our work provides initial evidence for a vital part of this vicious circle: circadian modulation of electrophysiology is indeed impaired in pathological tissue in humans in vivo.

A limitation of our study is the coverage of electrodes in each patient. As the implantation of the iEEG is to inform surgery, electrodes are typically confined to a single hemisphere or a series of neighbouring or connected regions. This means that each patient does not have complete data across all brain areas, and the AUC values calculated are based only implanted regions. A related limitation is that pathology might be present more broadly across the brain than the implanted region or SOZ[37–39]. It is also possible that the typical chronobiological rhythms of patients could be disrupted by surgery and implantation of the iEEG electrodes, their stay in the epilepsy monitoring unit, and changes to anti-seizure medication. An additional limitation of this study is that this cohort may not be representative of people with epilepsy in general because they are only those for whom it has been appropriate to undergo epilepsy surgery as they have drug-refractory focal epilepsy. A related question is whether the diminished rhythms we see may be associated with specific genetic markers of epilepsy. Future work should seek to investigate the whether specific genetic markers may help to explain the strongly diminished rhythms seen in some individuals.

Future work will also seek to establish a normative map[40–43] of the expected power of each ultradian and circadian rhythms in healthy tissue. Such a map would allow a better estimation of whether any particular observed rhythm was abnormal in a given brain region. To create such a normative map, we need to draw on a large and diverse cohort of subjects and assess their healthy tissue to estimate the expected power. This approach has previously shown success in the context of intracranial EEG and epilepsy[42,43], but was limited to assess short-term signal properties. Future work should extend this line of work into ultradian and circadian rhythms.

Finally, a key limitation was our analysis of the ultradian rhythms. Due to the lack of a central pacemaker, ultradian rhythms are extremely variable between subjects, and possibly also over longer time periods. Our coarse grouping of these rhythms is inspired by literature, but not data-driven. Our results in this context should thus be interpreted as a lack of fluctuations on timescales below one day in pathological tissue, rather than any specific rhythmic activity being diminished. Future work should investigate these fluctuations from the perspective of episodic, but most likely not strictly rhythmic events[44].

In summary, pathological brain regions show weakened chronobiological rhythms of brain activity. At present, the causal direction of this association is not clear, but the effect is independent of seizure occurrence and brain region. These findings encourage future research to consider chronobiological rhythms in the development of disease models and treatments.

## Methods

### Preprocessing of long-term iEEG recordings

We analysed long-term iEEG recordings from 38 subjects with refractory focal epilepsy from the National Hospital for Neurology and Neurosurgery (Table 1). Data was stored in the National Epilepsy & Neurology Database[45] and processed with approval of Newcastle University Ethics Committee (42569/2023). All subjects consented to the use of their data.

For each subject we processed their entire available iEEG recordings. Firstly, we divided each subject's iEEG data into 30 s non-overlapping, consecutive time segments. All channels in each time segment were re-referenced to a common average reference. In each time segment, we excluded any noisy channels (with outlier amplitude ranges) from the computed common average. To remove power line noise, each time segment was notch filtered at 50 Hz. Finally, segments were band-pass filtered from 0.5 to 80 Hz using a 4th order zero-phase Butterworth filter (second order forward and backward filter applied) and further downsampled to 200 Hz. Missing data were not tolerated in any time segment and denoted as missing for the downstream analysis.

We then calculated the iEEG band power for each 30 s time segment for all channels. We extracted iEEG band power from 30 s non-overlapping iEEG segments in five frequency bands ($\delta$: 1–4 Hz, $\theta$: 4–8 Hz, $\alpha$: 8–13 Hz, $\beta$: 13–30 Hz and $\gamma$: 30–47.5 Hz, 52.5–57.5 Hz and 62.5–77.5 Hz) using Welch's method with 3 s non-overlapping windows. In detail, for each channel in every 2 s window we calculated the power spectral density (PSD) and used Simpson's rule to obtain the band power values which then averaged over all time windows within a 30 s segment to get the final band power values. In order to remove electrical noise, we selected custom range limits for the gamma frequency band. We $log_{10}$-transformed and normalised the band power values to sum to one for each 30 s segment. We then averaged over the relative log band power of all electrodes included in each ROI, thus obtaining, for each subject, one matrix of relative log band power at the ROI level for each frequency band (of size number of ROIs by number of 30 s segments).

In the majority of resulting relative band power matrices had missing data, which we imputed (Suppl. S1) to enable extraction of rhythms and their time-varying characteristics, such as instantaneous frequency and amplitude. Imputed data were not used for subsequent analysis, and segments with imputed data were blanked after extraction of rhythms.

### Extracting chronobiological rhythms using bandpass filter

To extract rhythms of various timescales from the relative log band power time series, we performed a 4th order zero-phase Butterworth filter (second order forward and backward). Within each subject, we extracted ultradian rhythms in different period bands (1–3h, 3–6h, 6–9h, 9–12h, 12–9h). Finally, we denoted a circadian rhythm with period length of 19h-1.3d. Future work may consider wavelet-based methods, or empirical mode decomposition methods for this step, but for simplicity and ease of adoption, we opted for a simple filter in this study.

### MRI processing for identifying regions and resected tissue

To map electrode coordinates to brain regions we used the same methods as described previously[35]. In brief, we assigned electrodes to one of 128 regions from the Lausanne scale60 atlas[46]. We used FreeSurfer to generate volumetric parcellations of each patient's pre-operative MRI[46,47]. Each electrode contact was assigned to the closest grey matter volumetric region within 5 mm. If the closest grey matter region was >5 mm away then the contact was excluded from further analysis.

To identify which regions were later resected, we used previously described methods[42,48]. We registered post-operative MRI to the pre-operative MRI and manually delineated the resection cavity. This manual delineation accounted for post-operative brain shift and sagging into the resection cavity. Electrode contacts within 5mm of the resection were assigned as resected. Regions with > 25% of their electrode contacts removed were considered as resected for downstream analysis.

### Determining if chronobiological rhythms are diminished in pathological tissue

In order to quantify the strength of each band power chronobiological rhythm at the ROI level for each subject, we computed the average power of each rhythm obtained from the bandpass filtered signal. In order to compare the strength of each rhythm between pathological and healthy tissue, we computed the area under the receiver operating curve (AUC). AUC values >0.5 indicate diminished band power rhythms in pathological tissue, while AUCs <0.5 indicate diminished band power rhythms in spared ROIs and finally AUC = 0.5 indicates no discrimination between pathological and healthy tissue. To test whether a distribution of AUC values was significantly >0.5 we used a one-sided Wilcoxon rank-sum test, and indeed the AUC is a normalised version of the Wilcoxon rank-sum test statistic.

We used the clinically identified regions where seizures originate (seizure onset zone, or SOZ) as a proxy for pathological tissue. To ensure robustness of our results, we also used the alternative definition of the tissue that was subsequently surgically removed, as it was deemed central to generating epileptic seizures. We obtained similar results (Suppl. S5) with either definition, and will use the broad term of "pathological tissue" throughout the paper to refer to tissue that is most likely functionally (causing seizures) and/or structurally pathological.

### Fitting a mixed effects model to control for brain area

To show that pathological tissue predicted diminished rhythms of brain activity above any brain area specific effect, we fit a mixed effects model predicting the rhythm power in a given ROI from a categorical variable representing lobe within which the ROI resides, and whether that ROI was pathological, grouping by a random patient offset. Equation (1) describes the model, where $RhythmPower_{i,j}$ is the power of the rhythm for patient $i$ in ROI $j$, $\beta_0$ is the fixed intercept, $\beta_1$ is the fixed

effect for pathological ROI, $hasPath_{i,j}$ is a binary variable indicating that ROI $j$ of patient $i$ is pathological, $\beta_{2..7}$ is the fixed effect of the region being in each lobe – a categorical variable with 7 levels corresponding to the x axis of Fig. 3a, $Lobe_{i,j}$ indicates whether ROI $j$ of patient $i$ is in each lobe, $u_i$ is the random intercept for patient $i$, and $\mathcal{E}_{i,j}$ is the error. Regions were grouped into 7 possible lobes/areas: frontal, temporal, parietal, occipital, cingulate, hippocampus and amygdala. The dependent variable (rhythm power) was standardised before fitting to give partially standardised coefficients, allowing us to interpret $\beta_1$ as the expected effect of pathology on rhythm power (in standard deviations)[49]. A likelihood ratio test was used to assess whether the effect of pathology was statistically significant – comparing to a reduced model without pathology as a variable. Maximum likelihood estimation was used in the fitting of models, with Python (3.8.10) alongside the statsmodels library (0.14.0) and its MixedLM function providing the software.

$$RhythmPower_{i,j} = \beta_0 + \beta_1 hasPath_{i,j} + \beta_{2..7}Lobe_{i,j} + u_i + \mathcal{E}_{i,j} \quad (1)$$

### Reporting summary

Further information on research design is available in the Nature Portfolio Reporting Summary linked to this article.

## Data availability

Source data are provided with this paper. Data used is available at: https://zenodo.org/record/8289342. Source data are provided with this paper.

## Code availability

All analysis was performed using Python (version 3.8) and MATLAB (version R2023a). Code used is available at: https://github.com/cnnp-lab/DiminishedRhythmsPathology.

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

## Acknowledgements

We thank members of the Computational Neurology, Neuroscience & Psychiatry Lab (www.cnnp-lab.com) for discussions on the analysis and manuscript; P.N.T. and Y.W. are both supported by UKRI Future Leaders Fellowships (MR/T04294X/1, MR/V026569/1). J.S.D. and J.d.T. are supported by the NIHR UCLH/UCL Biomedical Research Centre.

## Author contributions

Conceptualisation: M.P., P.N.T. and Y.W. Methodology: M.P., C.T., B.C.S. and Y.W. Software/validation: C.T., M.P. and Y.W. Formal analysis: C.T., M.P. and Y.W. Resources: F.A.C., B.D., J.S.D., A.Mc. and A.M. Data curation: M.P., C.T., S.J.G., G.B., Y.W., F.A.C., B.D., J.S.D. A.Mc., A.M. and J.d.T. Writing: C.T., M.P., P.N.T., Y.W. and J.d.T. Supervision: P.N.T. and Y.W.

## Competing interests

The authors declare no competing interests.
