## [Peer Review File · Nature Communications]

Diminished circadian and ultradian rhythms of human brain activity in pathological tissue in vivoREVIEWER COMMENTS

Reviewer #1 (Remarks to the Author):

In this manuscript, Panagiotopoulou and colleagues report the outcomes of an analysis of a unique dataset arising from long-term recordings of continuously intracranially recorded EEG from people suffering from epilepsy. The authors apply sophisticated robust analytical tools to this dataset and show that daily or circadian rhythms as well as less than 24h or ultradian rhythms in brain activity are diminished in epilepsy. Importantly, they show that such changes are not attributable to a reduction in the magnitude of the raw EEG signal so these changes appear bona fide. Interestingly, the severity of the pathology is independent of these changes in circadian and ultradian rhythms. Similarly, the reduction in rhythms is sustained over time and not dependent on seizure occurrence or brain area. Thus, this report provides one the first and fullest accounts from high quality recordings of human EEG of diminished circadian and ultradian rhythms in epilepsy that are not dependent on time, pathology, or brain area. This is an important study, but there are some points for the authors to consider:

- 1) It would be useful for the reader to know why a range of 19-31h was used to classify a rhythm as circadian--typically this is more closely associated with 20-26h, so this is a broader range. Similarly, in Methods 2.5, why was 36h used? It would be useful to have a fuller account of this decision making.
- 2) The authors used the term circadian but in essence they are recording from people who are subject to light-dark and other recurring time of day signals. Thus, they could replace circadian with 'daily' or another term as circadian is conventionally used to describe a rhythm or process that is sustained in constant environmental conditions (ie without time of day signals).
- 3) The authors conclude that they see no evidence of disrupted rhythms causing seizures, but later in the Discussion (bottom of page 12) they state disrupted rhythms exacerbate disease symptoms/progression (or perhaps even severity?). I think they are probably correct in the latter assertion, but the initial statement seems contrary to this. Could they please clarify what they mean?

Reviewer #2 (Remarks to the Author):

This manuscript by Panagiotopoulou and Thornton, et al reports on a very interesting study using long-term, continuous intracranially recorded EEG from patients with epilepsy. The authors pulled out interesting circadian and ultradian rhythms from the long-term recording and was able to characterize functional differences in 'pathologic' tissue or epileptic foci vs non-pathological tissue. This is very novel finding indeed in the field of circadian biology and epilepsy as it is the first study to demonstrate using human intracranial electrophysiology that there is indeed dysfunction in circadian rhythm in the brain of patients with epilepsy. I find it very interesting as well that the study seems to find intrinsic deficiency of

these biological rhythms in the seizure foci of most of the patients. Additionally, it is interesting finding that the dysfunction of the biological rhythms appear to be independent of the seizure occurrence.

I have a few questions and points that I would like the authors to consider in their response to my review:

- The authors had assigned the label of 'pathological tissue' to the epileptic foci or source or alternatively the tissue that was surgically removed by the surgeons as this was considered by the clinicians as necessary to remove for the benefit of the patients. However, as we know, epileptic surgery in itself is not always successful in achieving the seizure or cognitive outcome that the procedure is intended for. Additionally, there is increasing thought and research in the epilepsy field that suggests the perilesional area of the seizure foci may also be dysfunctional and therefore be 'pathological'. This does not change the results or interpretation of the authors; which I think is wonderful; but I wonder if calling it 'pathological tissue' can be an overgeneralization. I would personally label them as 'epileptic foci' rather than 'pathological tissue'.

- It sounds like from supplementary data that the authors had considered many other things; i.e: that the data is not only relevant to the delta power band and the deficiency is found across all rhythms; that there is no sex or age difference; - while I appreciate these all can be found on the supplementary data, on reading the manuscript, I did wonder about all these on first pass and would recommend the author include a sentence or two in the main manuscript to suggest that these were considered and had been analysed.

- One other main point for me was the lack of characterization of the epilepsy patient cohort themselves. Does the author have access to the clinical diagnosis of the patients? Are they TLE or genetic epilepsies for instance? If genetic, is there any genetic information available? Additionally, were the patients on any anti-epileptic medications prior to the surgery? These are all typically reported for best practice. Have the authors checked that their finding cannot be attributed to heterogeneity in antiepileptic medication in the patient cohort? Finally, I wonder if the authors have considered if this particular finding can be generalized to 'epilepsy' or even pathological in general when the dataset is all from drug-refractory focal epilepsy and thus bias towards this specific patient population.

- Regarding the seizure analysis, it is a very interesting finding that the diminished rhythm appear independent of seizure occurrence. Is the seizure in these patients inherently circadian themselves as has been reported in some other studies that it can be? Have the authors checked? Also, encouraging that the authors checked for the peri-ictal and inter-ictal periods; have the authors checked the post-ictal period where there is typically interesting silencing of the electrical rhythms in the brain?

- Have the authors considered if these reduction of rhythms can be predictive of the area of seizure foci and therefore in the author's suggestion an area of 'pathology'? Perhaps, there is another utility here in that this can be a biomarker of seizure foci and help the clinician identify area of pathology that is not inherent? Have the authors done the statistical analysis or modeling to check if it can be predictive?

Some minor comments:

In the introduction: 'Associations between altered epilepsy, Parkinson's, and dementia have been (as opposed to are; would be better, in my opinion) reported'

In the discussion:

'Our work provides initial evidence of a link between the strength of chronobiological rhythms and the healthy functioning of brain tissue in humans in vivo' - in my opinion, this is too strong of a leap or

inference from this study result.

'In our data, we observed no correlation recurrent seizures after surgery per se' - quite a confusing sentence, would suggest rephrase.

Overall, this is a very interesting and novel study. I hope that the above comments help push the authors to consider some points and build on this already great manuscript to put together a final version that will be impactful in the epilepsy and circadian field.

Reviewer #3 (Remarks to the Author):

NCOMMS-23-41021

Mariella Panagiotopoulou and colleagues explore daily rhythms in continuous, intracranial EEG recordings collected over several days in patients with seizure activity. This is a really exciting data set which likely yields provocative findings.

The authors quantify rhythmicity of EEG recordings by means of an AUC method where area under the curve measures of wave activity were compared between pathological tissue and control brain regions. This strikes me as a rather crude measure to characterize curve characteristics. I wonder why the authors have chosen that particular method, please provide the more detailed rationale supported by literature references. The daily fluctuations as presented in Figure 1 for delta power should be analyzed for amplitude and phase by methods established in the circadian field such as <https://www.ncbi.nlm.nih.gov/pmc/articles/PMC3119870/> which is also being used to explore clinical data streams. Given the multi day recordings another method to consider is wavelet transforms.

The patient cohort should be characterized in depth for their seizure phenotype. Were time of day preference of seizure activity observed in patients? Do data exist to determine entrainment of the patients, for example sleep wake rhythms, sleep onset as phase marker? I was surprised that the circadian rhythmicity was defined with a period of 19 to 31 hours. Since the patients were entrained by behavioral and environmental cues to a 24 hour day, wouldn't we expect to find rhythms with the period close to 24 hours? Is it known to what extent EEG outputs are driven by circadian, behavioral, and environmental components? Unclear is the temporal relationship between EEG recordings and surgical intervention.

I am missing approaches to demonstrate internal consistency for this data set. An idea could be, for example, to determine curve characteristics, amplitude, phase, on 24 hour increments and to compare bins with seizure activity versus bins without seizure activity on an intra individual basis, if possible. I would assume that seizure activity affects curve characteristics, for example, loss of amplitude, possibly a shift in phase. Another avenue to consider might be to factor in disease severity and association to

rhythm characteristics, a limitation might be of course the small sample size. Such approaches would strengthen this work.

For the introduction, I recommend summarizing the existing literature pertaining to EEG recordings, the current state of rhythmic analysis approaches and implications to phenotype epilepsy and advance therapeutic approaches. How would the authors synthesize this into a hypothesis for the paper? This could give the paper guidance, for example, could clinically actionable insight be extracted. In the title and throughout the paper, the term “rhythms” should be specified as “rhythms of ...”

Figure 1 a suggests an amplitude loss over time for both the control and pathological tissues. How does this compare to the other patients? Is there a loss of connectivity between brain tissue and electrode to be considered? There seems to be an edge effect, small amplitudes on day one and day 7 compared to days two through 6. Again, how does this compare to other patients?

Response to Reviewers' comments

We would like to thank all reviewers for their encouraging comments and acknowledging the importance of our work – these have been particularly well-received by the early-career lead-authors on this paper. We are also very grateful for their effort to help us improve our paper with their detailed suggestions. We will address each point in turn below.

Reviewer Requests:

Reviewer #1 (Remarks to the Author):

In this manuscript, Panagiotopoulou and colleagues report the outcomes of an analysis of a unique dataset arising from long-term recordings of continuously intracranially recorded EEG from people suffering from epilepsy. The authors apply sophisticated robust analytical tools to this dataset and show that daily or circadian rhythms as well as less than 24h or ultradian rhythms in brain activity are diminished in epilepsy. Importantly, they show that such changes are not attributable to a reduction in the magnitude of the raw EEG signal so these changes appear bona fide. Interestingly, the severity of the pathology is independent of these changes in circadian and ultradian rhythms. Similarly, the reduction in rhythms is sustained over time and not dependent on seizure occurrence or brain area. Thus, this report provides one the first and fullest accounts from high quality recordings of human EEG of diminished circadian and ultradian rhythms in epilepsy that are not dependent on time, pathology, or brain area. This is an important study, but there are some points for the authors to consider:

1.1) It would be useful for the reader to know why a range of 19-31h was used to classify a rhythm as circadian--typically this is more closely associated with 20-26h, so this is a broader range. Similarly, in Methods 2.5, why was 36h used? It would be useful to have a fuller account of this decision making.

We chose a broad range to allow for more irregular sleeping/schedules on the Epilepsy monitoring ward, as patients adjust to the invasive implantation after anaesthesia. However, we have rerun our analysis with the proposed narrower range of 20-26h. The results of this analysis are shown in supplementary section 8. They indicate that the results are broadly the same when using the narrower definition, and we have included this in our main text: "Applying a more narrow definition of circadian rhythms (20-26 hours) did not substantially alter the distribution of AUCs".

In section 2.5 we use a window equal to the rhythm period multiplied by 1.5. We have updated the text here to make this clearer: "we calculated a rolling median of the AUC using a window proportional to the rhythm period multiplied by 1.5 (e.g. 36 hours for the circadian rhythm)".

1.2) The authors used the term circadian but in essence they are recording from people who are subject to light-dark and other recurring time of day signals. Thus, they could replace circadian with 'daily' or another term as circadian is conventionally used

to describe a rhythm or process that is sustained in constant environmental conditions (ie without time of day signals).

We agree that terminology can be confusing depending on the field and context. We have struck a compromise in our revisions by using the term “daily” where appropriate and defining the terms circadian and ultradian early on, specifically stating that we use these terms in a wider context, regardless of constant or changing environments. I.e. daily and circadian are interchangeable terms for this paper. The use of both terms means that this paper should be accessible and discoverable by a wide range for scientists from various backgrounds.

1.3) The authors conclude that they see no evidence of disrupted rhythms causing seizures, but later in the Discussion (bottom of page 12) they state disrupted rhythms exacerbate disease symptoms/progression (or perhaps even severity?). I think they are probably correct in the latter assertion, but the initial statement seems contrary to this. Could they please clarify what they mean?

We now clarified this confusing point in our discussion as follows:

“Another open question is if the electrophysiological circadian disruptions are a cause or consequence of the epileptogenic pathology. In our data, we observed no correlation between the resection of brain areas showing diminished rhythms and subsequent surgical outcome. This observation would suggest that sparing regions with diminished rhythms does not cause recurrent seizures after surgery *per se*. Similarly, we observed no temporal correlation between the magnitude of diminished rhythms and seizure occurrence. These observations should be validated in a larger cohort in future, ideally with more complete spatial sampling. We can nevertheless conclude that we see no evidence of disrupted rhythms causing seizures directly or acutely in our cohort. This, however, does not rule out indirect effects of disrupted rhythms over longer timescale, where disrupted rhythms impact sleep or other restorative processes over days and weeks and as a consequence provoke or at least enable seizures.”

Reviewer #2 (Remarks to the Author):

This manuscript by Panagiotopoulou and Thornton, et al reports on a very interesting study using long-term, continuous intracranially recorded EEG from patients with epilepsy. The authors pulled out interesting circadian and ultradian rhythms from the long-term recording and was able to characterize functional differences in 'pathologic' tissue or epileptic foci vs non-pathological tissue. This is very novel finding indeed in the field of circadian biology and epilepsy as it is the first study to demonstrate using human intracranial electrophysiology that there is indeed dysfunction in circadian rhythm in the brain of patients with epilepsy. I find it very interesting as well that the study seems to find intrinsic deficiency of these biological rhythms in the seizure foci of most of the patients. Additionally, it is interesting finding that the dysfunction of the biological rhythms appear to be independent of the seizure occurrence.

I have a few questions and points that I would like the authors to consider in their response to my review:

2.1) The authors had assigned the label of 'pathological tissue' to the epileptic foci or source or alternatively the tissue that was surgically removed by the surgeons as this was considered by the clinicians as necessary to remove for the benefit of the patients. However, as we know, epileptic surgery in itself is not always successful in achieving the seizure or cognitive outcome that the procedure is intended for. Additionally, there is increasing thought and research in the epilepsy field that suggests the perilesional area of the seizure foci may also be dysfunctional and therefore be 'pathological'. This does not change the results or interpretation of the authors; which I think is wonderful; but I wonder if calling it 'pathological tissue' can be an overgeneralization. I would personally label them as 'epileptic foci' rather than 'pathological tissue'.

We agree that we should ideally use a different term to reflect the subtleties of the tissue, pathology and its role in epilepsy. The most precise description we believe is the following: the seizure onset regions (and the resected regions) approximate a region that is hypothesised to be necessary and sufficient to support seizures as part of a network and is often associated with some histopathological finding when removed. However, we cannot 100% verify the hypothesis in our cohort: even in a patient that is seizure-free after the removal of those regions, the patient may remain on anti-seizure medication and/or may develop seizures again months/years later. We also agree that focal epilepsy is regarded as a network disorder, with evidence of epileptogenic networks, and with post-surgical remission relating to interruption of the networks. Hence the term of the “epileptic focuc/foci” may be equally misleading here.

To ensure that this paper is accessible to a wide audience that may not have a background in epilepsy, we opted to, instead, explain our term better in the text upon first use:

“Clinically identified regions where seizures originate (seizure onset zone, or SOZ) are often used to approximate areas that are hypothesised to cause or support seizures as part of a network. These regions are often also associated with some histopathological finding when resected. We therefore propose to use SOZ regions as a proxy for “pathological tissue”. To ensure robustness of our results, we also used the alternative definition of the tissue that was subsequently surgically removed, as it was deemed central to generating epileptic seizures. We obtained similar results (Suppl. S5) with either definition, and will use the broad term of “pathological tissue” throughout the paper to refer to tissue that is most likely functionally (causing seizures) and/or structurally pathological.”

2.2) It sounds like from supplementary data that the authors had considered many other things; i.e: that the data is not only relevant to the delta power band and the deficiency is found across all rhythms; that there is no sex or age difference; - while I appreciate these all can be found on the supplementary data, on reading the manuscript, I did wonder about all these on first pass and would recommend the author include a sentence or two in the main manuscript to suggest that these were considered and had been analysed.

We have now added the references to all supplementary sections throughout the reporting of related results. For example, final paragraph of 2.2 now contains a few sentences describing the supplementary results on age and sex differences,

alternative definition of circadian (20-26 hr) and patterns of cycle amplitude across time. Likewise, the final paragraph of section 2.3 includes reference to supplementary work on other EEG bands and investigations of the raw signal power.

e.g. “Applying a more narrow definition of circadian rhythms (20-26 hours) did not substantially alter the distribution of AUCs (median AUC = 0.6) - shown in supplementary figure S8.1. In the example subject we see lower cycle amplitude at the start and end of the recording, a feature present in a minority of cases across the cohort. Patterns of cycle amplitude across the recording are explored in supplementary section S11.”

2.3) One other main point for me was the lack of characterization of the epilepsy patient cohort themselves. Does the author have access to the clinical diagnosis of the patients? Are they TLE or genetic epilepsies for instance? If genetic, is there any genetic information available? Additionally, were the patients on any anti-epileptic medications prior to the surgery? These are all typically reported for best practice. Have the authors checked that their finding cannot be attributed to heterogeneity in antiepileptic medication in the patient cohort? Finally, I wonder if the authors have considered if this particular finding can be generalized to 'epilepsy' or even pathological in general when the dataset is all from drug-refractory focal epilepsy and thus bias towards this specific patient population.

We have now included data on epilepsy type (TLE or extra TLE), sex, and anti-seizure medication (ASM) used prior to surgery. Supplementary section S7 now shows these results.

Our results indicate no sex differences, but a tentative difference between TLE and extra TLE.

Regarding medication: Because of the small cohort size and large variations in medication type and combinations we were not able to draw any strong conclusions from this, but we consider this an important question and are working on exploring the associations between ASM use and biological rhythms for future publication.

We did not have genetic data for these patients, so it was not possible to include this. We agree, however, that future work should consider this, and included this in Discussion: “A related question whether the diminished rhythms we see may be associated with specific genetic markers of epilepsy. Future work should also seek to investigate whether specific genetic markers may help to explain the strongly diminished rhythms are seen in some individuals”.

The final point regarding generalisability of our results has also been added to our Discussion in the limitations section: “An additional limitation of this study is that this cohort may not be representative of people with epilepsy in general because they are only those for whom it has been appropriate to undergo epilepsy surgery as they have drug-refractory focal epilepsy”.

2.4) Regarding the seizure analysis, it is a very interesting finding that the diminished rhythm appear independent of seizure occurrence. Is the seizure in these patients inherently circadian themselves as has been reported in some other studies that it can be? Have the authors checked? Also, encouraging that the authors checked for the

peri-ictal and inter-ictal periods; have the authors checked the post-ictal period where there is typically interesting silencing of the electrical rhythms in the brain?

The inherent circadian phase preference of seizure is an interesting point to explore further. To investigate whether the seizures themselves are inherently circadian in their occurrence in our cohort, we have calculated phase locking values for each patient, indicating the extent to which their seizures have a phase preference. While this was not evident in most individuals, a minority of 21 % do, in line with previous publications on similar patient cohorts [Leguia 2021 JAMA Neurol]. Supplementary section S9, presents this as well as a visual inspection of the 6 individuals with the greatest circadian phase preference in seizure timing. We found that in subjects who show phase preference, there was no deviation in their rolling AUC values at this phase.

Re post-ictal period: In supplementary section S10 we now show the distribution of AUC values calculated using only the post-ictal period, and comparing it to interictal data. We show that there is no significant difference between the postictal rolling AUC value and the interictal period.

2.5) Have the authors considered if these reduction of rhythms can be predictive of the area of seizure foci and therefore in the author's suggestion an area of 'pathology'? Perhaps, there is another utility here in that this can be a biomarker of seizure foci and help the clinician identify area of pathology that is not inherent? Have the authors done the statistical analysis or modeling to check if it can be predictive?

We do not believe the reduction in biological rhythms seen in the SOZ is consistent enough from patient to patient to be used as a predictive marker. While the AUC distributions show that most patients have a reduced rhythm in the SOZ, there is too much heterogeneity for this to be a good marker. For example, an AUC of 0.6 (median for circadian delta) is not generally considered good enough as a biomarker. In other words, our results highlight a biological group-level effect of diminished rhythms, but individuals clearly show variations that are modulated by additional factors.

Some minor comments:

In the introduction: 'Associations between altered epilepsy, Parkinson's, and dementia have been (as opposed to are; would be better, in my opinion) reported'

We have made the suggested changes.

In the discussion: 'Our work provides initial evidence of a link between the strength of chronobiological rhythms and the healthy functioning of brain tissue in humans in vivo' - in my opinion, this is too strong of a leap or inference from this study result. 'In our data, we observed no correlation recurrent seizures after surgery per se' - quite a confusing sentence, would suggest rephrase.

Changed to: "Our work provides initial evidence of an association between the strength of chronobiological rhythms and the healthy functioning of brain tissue in humans in vivo"

Overall, this is a very interesting and novel study. I hope that the above comments help push the authors to consider some points and build on this already great manuscript to put together a final version that will be impactful in the epilepsy and circadian field.

Changed to “In our data, we observed no correlation between the resection of brain areas showing diminished rhythms and subsequent surgical outcome.”

Reviewer #3 (Remarks to the Author): NCOMMS-23-41021

Mariella Panagiotopoulou and colleagues explore daily rhythms in continuous, intracranial EEG recordings collected over several days in patients with seizure activity. This is a really exciting data set which likely yields provocative findings.

3.1) The authors quantify rhythmicity of EEG recordings by means of an AUC method where area under the curve measures of wave activity were compared between pathological tissue and control brain regions. This strikes me as a rather crude measure to characterize curve characteristics. I wonder why the authors have chosen that particular method, please provide the more detailed rationale supported by literature references. The daily fluctuations as presented in Figure 1 for delta power should be analyzed for amplitude and phase by methods established in the circadian field such as <https://www.ncbi.nlm.nih.gov/pmc/articles/PMC3119870/> which is also being used to explore clinical data streams. Given the multi day recordings another method to consider is wavelet transforms.

Quantifying rhythmicity of EEG band power: Here, we have used a bandpass filter to isolate the components of the band power signal with a period corresponding to the biological rhythms of interest. As our signal is densely sampled, well-established time-frequency analysis methods can be applied, and we do not need to resort to statistical methods that were developed for sparsely-sampled time series (as the highlighted paper). We agree that some wavelet-based methods would have also been adequate, but opted for the simpler Fourier-based bandpass filtering method to allow easier adoption and comparison by future work. We did not use area under the time series as a measure. We have now clarified these points in Methods and Results.

AUC statistic: We use the AUC – area under the receiver operating characteristic curve – as a well-established measure to quantify the difference between the power of rhythms in the pathological brain areas vs. the remaining brain areas. The AUC statistic is also equivalent to the normalised Wilcoxon rank sum statistic, both measuring effect size between two distributions in a non-parametric manner. We have now clarified this in Methods.

Amplitude vs phase: In this work, we wanted to characterise only differences in the strength of rhythms, or in other words, the power in certain time scales. Investigations of phase differences is also of interest but not within the scope of this paper, and more complex for signals that may not be rhythmically generated (such as ultradian time scales).

3.2) The patient cohort should be characterized in depth for their seizure phenotype. Were time of day preference of seizure activity observed in patients? Do data exist to determine entrainment of the patients, for example sleep wake rhythms, sleep onset as phase marker? I was surprised that the circadian rhythmicity was defined with a period of 19 to 31 hours. Since the patients were entrained by behavioral and environmental cues to a 24 hour day, wouldn't we expect to find rhythms with the period close to 24 hours? Is it known to what extent EEG outputs are driven by circadian, behavioral, and environmental components? Unclear is the temporal relationship between EEG recordings and surgical intervention.

Temporal relationship of diminished rhythms to seizures: To address this point, we have now characterised the phase preference of seizures for each patient to the 24 daily cycle and their own circadian cycle in delta power (both in SOZ and remaining regions), presented in supplementary section S9. We found that for most patients there was no clear phase preference (median phase locking value around 0.4), in line with previous publication [Leguia 2021 JAMA Neurol]. For the 6 patients who had a high PLV we have shown their AUC values averaged across days to get a 24 mean, alongside the timing of their seizures. Visual inspection indicates no obvious relationship.

Definition of circadian time scale: We have also now replicated our main analysis with a narrower definition of circadian rhythms – 20 – 26 hours, as suggested by reviewer 1. This is presented in supplementary section S8. We find a very similar distribution of AUC values to the broader definition (median=0.6) and in the example patient a similar pattern of signal power across brain regions. Our choice of a wide band for the circadian rhythm was influenced by the patients being in the epilepsy monitoring unit where light levels, sleeping times, noise, pain, and anxiety would contribute to disruption of their circadian rhythm.

EEG & circadian rhythm: We have expanded our Introduction to include papers that clearly detect circadian rhythms in a range of EEG features.

EEG & surgical timing: As we used intracranially recorded EEG data, the time of surgery to implant the electrodes always precedes the EEG recording and the recordings generally start 18-24 hours after implantation. The timing of subsequent surgical removal of suspected epileptogenic tissue is usually a number of weeks after the end of the EEG recording, but varies from patient to patient.

3.3) I am missing approaches to demonstrate internal consistency for this data set. An idea could be, for example, to determine curve characteristics, amplitude, phase, on 24 hour increments and to compare bins with seizure activity versus bins without seizure activity on an intra individual basis, if possible. I would assume that seizure activity affects curve characteristics, for example, loss of amplitude, possibly a shift in phase. Another avenue to consider might be to factor in disease severity and

association to rhythm characteristics, a limitation might be of course the small sample size. Such approaches would strengthen this work.

As detailed in the above comment we have added an exploration of phase-locking of seizures to the 24-hour cycle, as well as the patients' own daily cycle in delta power, both in the SOZ and other brain areas. Our data seizure phase locking results are very similar to those reported in the literature, thus demonstrating consistency of our data.

For those patients who did have a phase preference for their seizures, the AUC values showed no consistent deviation during the preferred seizure phase. This shows that our results have not been driven by a seizure-specific effect. A more general investigation of the causal effect of seizures on biological rhythms is of interest, but not within the scope of this paper due to its retrospective design. Future studies in animal models may shed light on this important question.

3.4) For the introduction, I recommend summarizing the existing literature pertaining to EEG recordings, the current state of rhythmic analysis approaches and implications to phenotype epilepsy and advance therapeutic approaches. How would the authors synthesize this into a hypothesis for the paper? This could give the paper guidance, for example, could clinically actionable insight be extracted.

We have now added additional material to the introduction to provide more background on the detection of circadian rhythms in EEG recordings, and current approaches for analysing biological rhythms. We also explicitly posed our hypotheses in Introduction as three related questions: "In the following sections we will examine (1) if circadian and ultradian rhythms are altered in pathological tissue, (2) if the alterations can be attributed to pathology alone, and (3) if any alterations are dependent on seizure occurrence."

In the title and throughout the paper, the term "rhythms" should be specified as "rhythms of"

We agree that being explicit about the modality of the recorded rhythm is important. However, in some sections, we are referring to rhythms of various modalities, or rhythmicity in general. We have now revised for specific modalities throughout the text where appropriate, including the title.

3.5) Figure 1 a suggests an amplitude loss over time for both the control and pathological tissues. How does this compare to the other patients? Is there a loss of connectivity between brain tissue and electrode to be considered? There seems to be an edge effect, small amplitudes on day one and day 7 compared to days two through 6. Again, how does this compare to other patients?

Patterns of biological rhythm power over the recording period are heterogeneous across the cohort. We have added supplementary section S11 to provide more detail on this. Here we have calculated the power of the rhythm over time for regions in the SOZ and in the rest of the brain, then assigned each patient into three categories.

Patients that show clear amplitude loss from the first couple of days to the rest of the recording (SOZ = 12, other = 10). Patients that show curves that fit well to a Gaussian, implying small amplitudes at the edges and higher in the middle (SOZ=13, other =15). Patients that fit neither of these categories, and show flatter curves (SOZ=14, other=14). There are several factors that may contribute to these patterns, including the anaesthesia used in surgery, changes to the impedance of the electrode contacts, changes in anti-seizure medication dose across the recording period, pain and other discomfort, and patient adaptation being on the epilepsy monitoring unit. Untangling these effects is a difficult challenge, but we are hoping to explore some of these, in particular drug tapering effects in future work.

Regardless, these effects are relatively global, affecting recording site in the SOZ and in the rest of the brain equally, and therefore do not impact our conclusions regarding pathological vs. non-pathological cerebral tissue.

REVIEWERS' COMMENTS

Reviewer #1 (Remarks to the Author):

In this study, the authors provide a sophisticated analysis to a unique set of recordings of brain activity with patients with epilepsy. This provides insight into how brain EEG (recorded intracellularly) can vary over circadian and ultradian time scales and can differ between patients and people without epilepsy.

The authors have done a good job responding to the concerns raised in the previous round. I have one very minor question--how did the authors accommodate for occasional failure in data acquisition? For example, if the recording for an hour or two became noisy?

Reviewer #2 (Remarks to the Author):

The authors have addressed all my comments from the previous review process. I applaud the authors for making the changes and including many additional analysis that hopefully enrich the interpretation of this dataset.

Reviewer #3 (Remarks to the Author):

Thank you for addressing the questions and concerns raised.

Note the type "Diminsihed" in Figure S 7.2